# Comparative Study on the Generation and Characteristics of Debris Induced by Fretting and Sliding

**DOI:** 10.3390/ma15124132

**Published:** 2022-06-10

**Authors:** Po Zhang, Zhaobing Cai, Wenjun Yang, Juan Chen, Shiyuan Luo, Liangcai Zeng

**Affiliations:** 1Hubei Key Laboratory of Mechanical Transmission and Manufacturing Engineering, School of Machinery and Automation, Wuhan University of Science and Technology, Wuhan 430081, China; caizhaobing@wust.edu.cn (Z.C.); yangwenjun@wust.edu.cn (W.Y.); chenjuanwhy@wust.edu.cn (J.C.); shiyuanluo@wust.edu.cn (S.L.); zengliangcai@wust.edu.cn (L.Z.); 2Key Laboratory of Metallurgical Equipment and Control Technology, Ministry of Education, Wuhan University of Science and Technology, Wuhan 430081, China

**Keywords:** wear debris, generation, characteristics, fretting wear, sliding wear

## Abstract

**Objectives:** The aim of the present work was to comparatively investigate the generation and characteristics of fretting and sliding wear debris produced by CuNiAl against 42CrMo4. **Methods:** Tribological tests were conducted employing a self-developed tribometer. Most experimental conditions were set the same except for the amplitudes and number of cycles. Morphological, chemical, microstructural and dimensional features of the worn area and debris were investigated using optical microscope (OM), X-ray diffraction (XRD), scanning electron microscopy (SEM) with energy dispersive spectroscopy (EDS) and a laser particle sizer. **Outcomes:** Not only wear scar profiles but also the wear debris color, distribution and generated amount under fretting and sliding wear modes were quite different, which can be attributed to the significant difference in wear mechanisms. Particle size analysis indicates that the fretting debris has a smaller size distribution range; the biggest detected fretting and sliding wear debris sizes were 141 μm and 355 μm, respectively. Both fretting and sliding debris are mainly composed of copper and its oxides, but the former shows a higher oxidation degree.

## 1. Introduction

Wear debris is generated by the rubbing of two surfaces. Its concentration, size, morphology and composition are associated with the wear mechanisms of surfaces [1,2]. Debris can be detected by offline methods such as ferrographs and spectrographs, as well as online methods including resistive-capacitive, acoustic, optical and inductive, which are useful for the condition monitoring of a wind turbine gearbox [3] and for remaining useful life prediction of aviation hydraulic pumps [4], etc.

For fretting and sliding wear debris, the formation mechanisms and micro-characteristics could be different. As fretting typically features a much smaller amplitude compared with the contact area width (Figure 1), most of the contact area keeps in contact as such, meaning some wear debris remains entrapped [5]. The retained debris undergoes repeated crushing, which affects the contact pressure distribution and changes the development of the fretting wear scar [6,7]. In sliding tests, though some debris is inevitably entrapped at the contact zone, most debris is expelled, leading to more direct metallic contact [8].

The generation mechanisms and micro-characteristics of fretting and sliding wear debris have attracted many researchers’ attention. Kirk et al. [9] investigated the formation and destruction of fretting wear debris beds, and found that the ejected debris particle size mainly varied with the displacement amplitude, largely independent of the number of cycles. Everitt et al. [10] identified the mechanical characteristics of the debris layer with a nano-indentation technique. They found that compared to the Ti-6Al-4V substrate, the compacted debris layer was about twice as hard, but Young’s modulus was only slightly different. Blades et al. [11] explored the types of fretting debris and the effect on the wear rate; they found that the particle size, rather than the hardness, can be better used to predict the wear rate. Soria et al. [12] studied the wear debris resulting from Incoloy 800 steam generator tubes fretting against AISI Type 304 stainless steel. They found that the debris is composed of agglomerations of nano-crystalline oxide particles with sizes ranging from 5 to 20 nm, and the crystal structure of debris depends on the displacement amplitude. Fan et al. [13] proposed an attenuation function to quantitatively depict the removal of debris in oil lubricated systems, and they verified this experimentally through two simulation tests. Nine et al. [14] reviewed the characterization and corresponding biological response of wear debris in artificial hip and knee joints, and they found that the geometrical characteristics of the wear particles are affected by the type, lubricant, material combination and geometry of the joints and bearing. Fischer et al. [15] surveyed the development, appearance and properties of debris generated by ceramics, and they found that the debris formed mostly by fractures on different scales: at low, intermediate and high loads, by microfracture, grain boundary fatigue and macroscopic fracture, respectively. 

However, our literature review showed that the debris generated by fretting and sliding were usually examined separately. Though some comparative studies exist [16,17], the wear debris was obtained with quite different tribometers or experimental parameters, thus reducing the comparative significance. For example, Laux et al. [18] researched the wear debris produced by polyaryletherketones under fretting and multi-directional sliding modes, with the contact pressure set at 50 Mpa and 5 Mpa, respectively.

In this study, morphological, chemical, microstructural and dimensional features of fretting and sliding debris were comparatively investigated. Rather than being generated by quite different testing rigs or under quite different test conditions, the two kinds of debris were generated on the same self-developed tribometer. The fretting and sliding experiments were carried out using the same material and contact configuration, under the same normal load and total relative angular displacement, but under different amplitudes and numbers of cycles. This research provides an intuitive insight into the generation and characteristics of debris produced under the two distinctly different modes.

## 2. Experimental Details

### 2.1. Materials and Specimens

Our previous studies [19,20] indicate that for the controllable pitch propeller (CPP), both fretting and sliding wear can occur on its blade bearing, depending on the pitch state (fixed or adjusted). In this research, CuNiAl and 42CrMo4, which are the materials that make up the blade bearing, were used to prepare the upper and lower specimens, respectively. For CuNiAl and 42CrMo4, their respective E moduli are 121,000 Mpa and 212,000 Mpa, Poisson’s ratios are 0.33 and 0.3 and the degrees of hardness (HB) are 127 and 220, respectively. The lower specimens were machined to be 40 mm × 30 mm × 7 mm blocks. The contact section of the upper specimens was designed to be two raised 45° sectors (Figure 2b). So, the contact area of the friction pair was a partial ring shape, which can help to increase the wear depth as well as improve the contact uniformity [21,22]. Before tests, the surfaces to be worn were firstly polished to *S_a_* = 0.5 μm, and then the specimens were cleaned ultrasonically in alcohol bath for 5 min.

### 2.2. Testing Rig and Parameters

The tribological tests were conducted using a self-developed testing rig [23]. The upper and lower specimens were fixed on the specimen holders. The friction pair was driven by a high-precision stepping motor with the resolution of 0.018°. The peak-to-peak stroke, i.e., the fretting and sliding amplitude, was controlled by setting the stepping motor parameter. The friction force was measured by a torque sensor (range: −2.5–+2.5 Nm; error: ≤0.5% F·S) in real-time. The normal load was applied as a dead weight. More details of the testing rig have been described in detail elsewhere [19,21,24].

As shown in Table 1, the angular displacement amplitude and number of cycles for the fretting experiments were set to be 15 times and 1/15 of the sliding tests, respectively, so for the two wear modes, the total angular displacement in each test was the same: 720,000°. The fretting and sliding frequency were set to 3 Hz and 0.2 Hz, respectively. Besides this, all the tests were conducted under the normal load of 106 N, at ambient temperature and under a dry condition. For each test condition, the experiment was repeated three times.

### 2.3. Analysis Methods

After tribological tests, optical characteristics of the worn areas and wear debris were firstly observed with a 3D digital microscope (DSX 510, Olympus Corp, Tokyo, Japan). Then, a higher-magnification investigation was carried out with a field-emission scanning electron microscope (FESEM, ThermoFisher/Apreo S HiVac, Waltham, MA, USA). Chemical characteristics of the wear debris were analyzed with energy dispersive X-ray spectroscopy (EDS, the appendant of SEM) and X-ray diffraction (XRD, Rigaku/SmartLab SE, Tokyo, Japan). The particle size distribution was determined by a laser scattering instrument (Mastersizer 2000, Malvern Instruments, Malvern, UK) with water as the dispersant; the detection range was from 0.02 to 2000 μm in an equivalent spherical diameter.

## 3. Results and Discussion

### 3.1. Optical Observation

After tribological tests, the worn surfaces on the lower specimens were firstly observed with a high-resolution 3D optical digital microscope (OLYMPUS, DSX 510). It can be seen that the fretting worn area is covered with a dark green debris layer, and surrounded by loose debris (Figure 3a). The fretting experiments were conducted under a dry condition and in a gross slip state, resulting in severe material removal and debris formation [25]. Yet, as the amplitude was very small, some of the debris got stuck in the contact zone and thus could not be fully oxidized, which created the dark green color. The sliding worn area looks bright, with the base material color exposed (Figure 3b). The debris mainly piles up at the ends of the arc contact area, indicating that it largely moved along the relative motion direction during the sliding process. Besides this, as it is easier to be expelled and oxidized, the sliding wear debris looks black. 

After removing the loosely bounded wear debris, 3D and 2D cross-sectional morphologies of the wear tracks were obtained using the 3D optical digital microscope. As the sliding worn surface was very big, image stitching was used to show the whole wear area in a picture. On the fretting worn surface, deep pits can be observed (Figure 4a), which are possibly formed by adhesion, cracks and delamination during the fretting process. Comparatively, the sliding wear trace is flatter than the fretting one (Figure 4b), and it is about 400 μm wider and 30 μm deeper. The fretting and sliding wear volumes were obtained using the 3D optical digital microscope with the following steps: (1) setting the reference plane by using the middle height of micro-convex bodies on the unworn area; (2) delineating the border of the worn area; (3) calculating the pit volume of the worn area below the reference plane. The fretting and sliding wear volumes were 7.144 × 10^8^ μm^3^ and 20.982 × 10^8^ μm^3^, respectively. Figure 5 presents the variation of friction torque and energy dissipation with angular displacement. This shows that the sliding friction torque ran into the steady state later than the fretting one; besides this, the steady friction torque of the sliding one was about 50% higher. The energy dissipation increased linearly with the angular displacement; the energy dissipation in the sliding test was about 40% higher than that in the fretting test. The above analysis indicates that compared to fretting tests, the wear volume increased more than the energy dissipation for the sliding tests. It can be inferred that wear rate of the sliding test was much higher than the fretting counterpart. This is because, without the protection of a thick debris layer, direct contact between metal asperities commonly occurs, leading to an increase in adhesive and abrasive wear. Moreover, according to our previous study [26], work hardening has a very serious impact on the fretting-worn surface, while it has a very slight impact on the sliding one; increasing the surface hardness is thus beneficial to increase the wear resistance. 

### 3.2. SEM Observation

Figure 6 presents typical SEM morphologies of the worn surfaces. Before ultrasonic cleaning, the fretting worn area is covered with a compacted debris layer. Consistent with the optical morphology in Figure 4, the wear scar is uneven and scatted with deep pits. According to the “third body concept” developed by Godet and co-workers [27,28], the fretting debris layer has several functions: firstly, supporting the load; secondly, participating in accommodation velocity; thirdly, avoiding direct interactions by separating the surfaces in contact. After ultrasonic cleaning, flake pits and cracks can be observed. The cracks are firstly nucleated below the fretting surface with highly compressive stress. Then, parallel to the surface, further deformation will lead to the extension and propagation of a crack at a certain depth. Finally, delamination of large flake pits occurs when the cracks propagate to the surface. On the sliding worn surface, grooves can be observed along the relative movement direction (Figure 6b). The thin debris layer scatters more evenly. After ultrasonic cleaning, the sliding worn surface mainly features abrasion traces. Comparatively, delamination and cracks cannot be easily observed. This is because compared with the initiation and propagation of a crack, the removal of material is dominant under a sliding condition [29].

SEM morphologies of the loose debris are displayed in Figure 7. During fretting tests, flaky wear debris peeled off and was repeatedly crushed and ground, mainly forming fine particles (Figure 7a). Under the sliding condition, the substrate material was mainly peeled off by abrasion, but part of the debris moved out of the contact area due to the large relative displacement amplitude. So, the sliding wear debris contained not only fine particles but also large flakes and chips. According to our previous study [26], the flaky and striped sliding wear debris looks brown to the naked eye, that is to say, the same as the base material of CuNiA. Yet, due to reflection, it is not easy to be identified under the microscope, as we can see in Figure 3.

### 3.3. Particle Size Analysis

The generated loose wear debris were collected and quantitatively analyzed by a laser particle size analyzer (Mastersizer 2000). The particle size distribution is graphically shown in Figure 8. This depicts that the two curves have similar evolution trends, but the fretting one shows a higher peak and smaller span, indicating that the fretting debris has a smaller size range. Quantitatively, compared to the sliding debris, d (0.1) of the fretting debris is bigger, while the d (0.5) and d (0.9) are smaller. Besides this, the biggest detected fretting and sliding wear debris sizes were 141 μm and 355 μm, respectively. From the above data, it can be inferred that there are more big particles in sliding wear debris.

### 3.4. Element and Phase Analyses

Figure 9 displays EDS patterns for the collected wear debris. This indicates that oxygen can be detected in both the fretting and sliding wear debris, meaning a violent oxidation reaction occurred during the wear process. According to previous studies [30,31], under dry fretting and sliding conditions, the rubbing interface had a high flash temperature, which promotes the formation of oxides [32,33]. Figure 9 indicates that oxygen in the fretting debris is 3% higher than that in the sliding debris. The oxides can reduce friction and wear effectively, and given the fact that the fretting debris is much thicker than the sliding one (Figure 3), this may be the reason why the friction torque of the fretting test is comparatively lower than the sliding one, as can be seen in Figure 5. The phase makeup of the wear debris has been analyzed by XRD, indicating that the debris is mainly composed of copper and its oxides (Figure 10).

## 4. Conclusions

In this paper, studies on the generation and characteristics of fretting and sliding wear debris were conducted with a self-developed tribometer. Morphological, chemical, microstructural and dimensional characteristics of the worn surfaces and collected wear debris were comparatively studied. The main conclusions are as follows:The generated sliding wear debris was about twice that of the fretting wear debris. This is because for a sliding worn surface, without the protection of a thick debris layer, direct metal-to-metal contact occurs, leading to an increase in adhesive and abrasive wear.The fretting debris was mainly formed by fine particles due to repeated crushing and grinding. The sliding debris contained not only fine particles but also large flakes and chips as part of the debris moved out of the contact area.Element and phase analyses indicated that a violent oxidation reaction occurred during both the fretting and sliding wear processes. The phase makeup of the fretting and sliding debris was mainly copper and its oxides.

## Figures and Tables

**Figure 1 materials-15-04132-f001:**
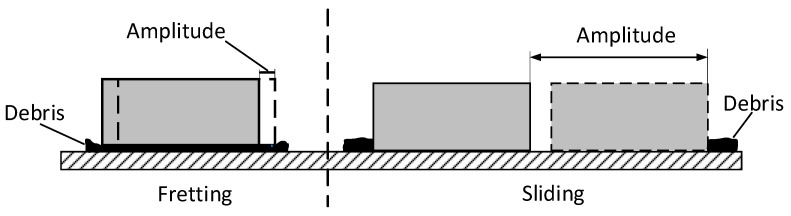
Concept of overlap in fretting and sliding.

**Figure 2 materials-15-04132-f002:**
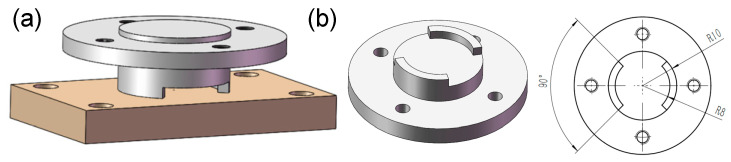
Schematic diagram of: (**a**) contact between the upper and lower specimens; (**b**) upper specimen.

**Figure 3 materials-15-04132-f003:**
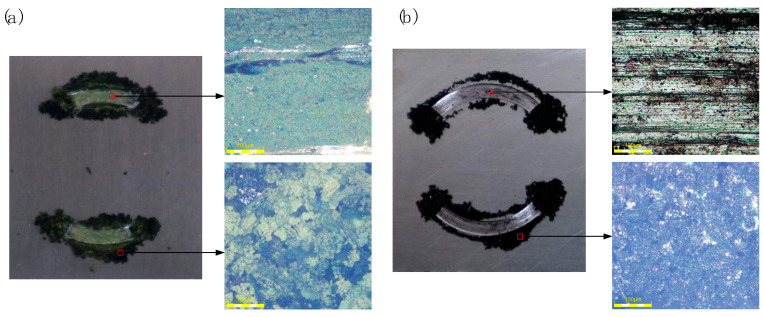
Optical observation of the worn area and debris: (**a**) fretting; (**b**) sliding.

**Figure 4 materials-15-04132-f004:**
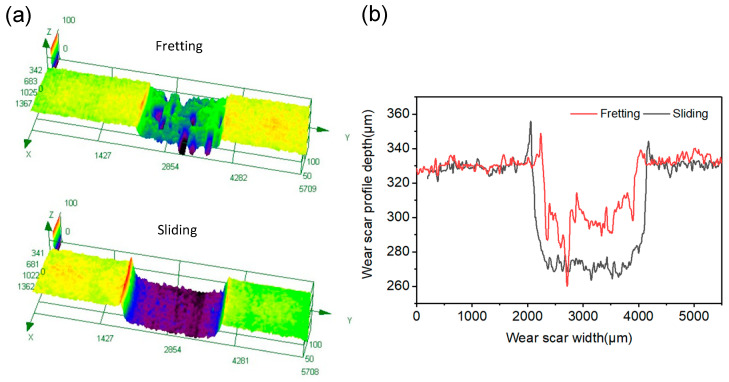
Cross-sectional profiles of the fretting and sliding worn traces: (**a**) 3D profiles; (**b**) 2D profiles.

**Figure 5 materials-15-04132-f005:**
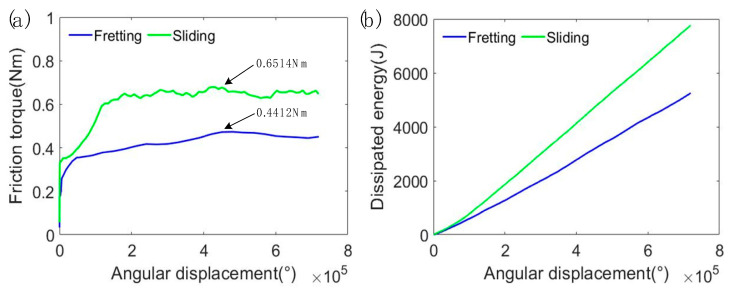
Variations of: (**a**) friction torque; (**b**) accumulated dissipated energy.

**Figure 6 materials-15-04132-f006:**
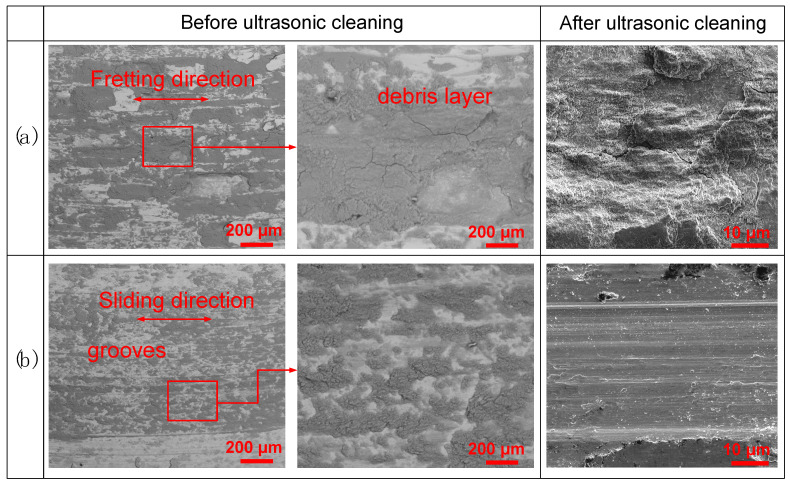
SEM morphologies of the worn surfaces: (**a**) fretting; (**b**) sliding.

**Figure 7 materials-15-04132-f007:**
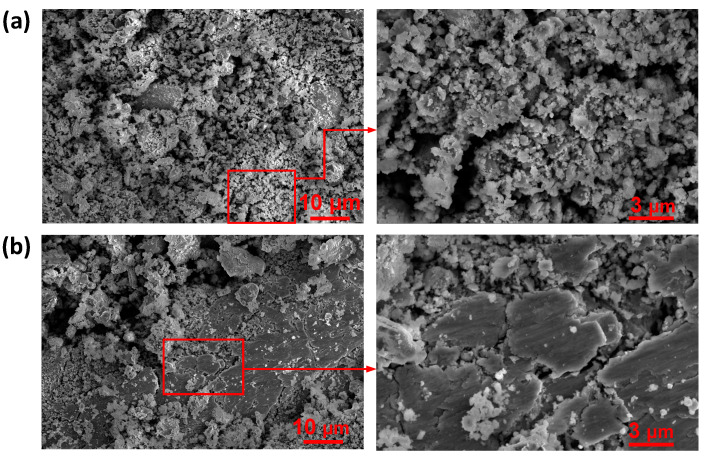
SEM morphologies of wear debris: (**a**) fretting; (**b**) sliding.

**Figure 8 materials-15-04132-f008:**
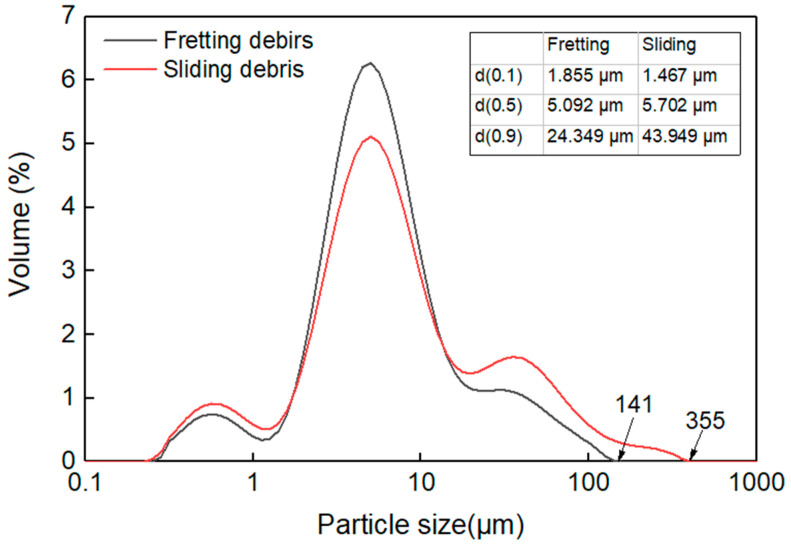
Particle size distribution of the wear debris.

**Figure 9 materials-15-04132-f009:**
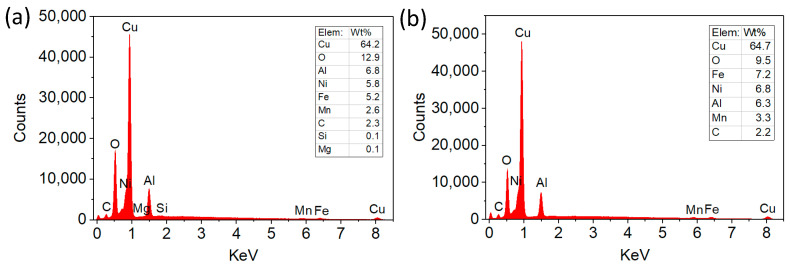
EDS patterns of the wear debris: (**a**) fretting; (**b**) sliding.

**Figure 10 materials-15-04132-f010:**
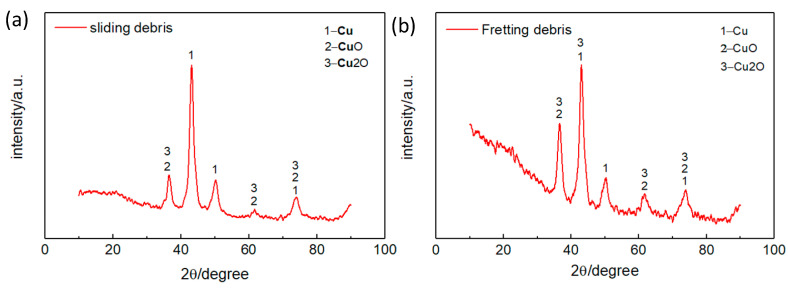
XRD pattern of: (**a**) fretting debris; (**b**) sliding debris.

**Table 1 materials-15-04132-t001:** Fretting and sliding test parameters.

	Load (N)	Amplitude (°)	Speed(°/s)	Number of Cycles	Total Angular Displacement (°)	Frequency (Hz)
Fretting	106	1.5	18	120,000	720,000	3
Sliding	106	22.5	18	8000	720,000	0.2

## Data Availability

The data presented in this study are available on request from the corresponding author.

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
