# Peer review of "Comparative Study on the Generation and Characteristics of Debris Induced by Fretting and Sliding"

_materials, 2022, doi:10.3390/ma15124132_

Round 1

Reviewer 1 Report

1.      Abstract need to be modified with objectives and outcomes.

2.      In the end of the introduction novelty of the work need to be highlighted.

3.      Method for calculating sliding wear need to include in the manuscript.

4.      Conclusion can be further elaborated.

5.      The paper should be proofread again, preferably by a technically qualified person or a professional technical editor.

6.      Add future scope of work before references

7.      Cite latest articles related to wear and lubrication example

Surya, M. S., & Gugulothu, S. K. (2022). Fabrication, mechanical and wear characterization of silicon carbide reinforced Aluminium 7075 metal matrix composite. Silicon, 14(5), 2023-2032.

Reviewer 2 Report

This paper shows a comparative analysis of the debris created by fretting and sliding.
The paper is quite clear and the distinction between the two types of debris is interesting.
However, in my opinion, some issues need to be considered:
1. Abstract. in the sentence: 'laser particle sizer, etc.' the use of 'etc.' is not recommended because all the types of investigations are listed.
2. 'by the large overlap between movement amplitude and contact area' this sentence is not clear, what is the intended meaning of the word overlap?
3. Fig. 2. Why was this shape of the indenter selected? A non-uniform distribution pressure could result along the two wide circular sides.
4. Table 1. In the table the sliding velocity and the load (that should be the same for both tests) would be useful to be reported.
5. Fig. 3. The role of the colours seems quite important in the discussion about the observation of the debris. Are they related to the oxidation level? However, maybe the colours are dependent on the optical system used for the observation.
6. 'The fretting and sliding wear volumes were..' how were these volumes measured?
7. The sentence 'with in fretting tests' seems incorrect.
8. In the paper it is stated that the detected biggest fretting and sliding wear debris sizes are 141 micron and 355 micron, respectively. How were these sizes detected?
9. 'given the fact that the fretting debris is much thicker than the sliding one (Fig. 3)' Why should this be evident from Fig. 3? The size distribution is shown in Fig. 5 and, if I understood well, the fretting one is more peaked, but mostly similar except the extreme values.
10. Fig. 10. Do not use spades diamonds or other similar symbols.
11. The experimental setup was already presented in Refs. [23, 24], and in particular in Ref. [24] the same materials were used. Thus the novelty of this paper should be well emphasized in the introduction.
Kind regards

Round 2

Reviewer 1 Report

The authors have incorporated all the comments in the manuscript and improved the quality of paper

Reviewer 2 Report

The paper was found properly revised and all the raised issues well addressed.
Kind regards